# Microstructure and Wear Resistance of Multi-Layer Ni-Based Alloy Cladding Coating on 316L SS under Different Laser Power

**DOI:** 10.3390/ma14040781

**Published:** 2021-02-07

**Authors:** Shaoxiang Qian, Yibo Dai, Yuhang Guo, Yongkang Zhang

**Affiliations:** 1College of Mechanical Engineering, Jiangsu University, Zhenjiang 212013, China; 2School of Modern Equipment Manufacturing, Zhenjiang College, Zhenjiang 212028, China; 3School of Materials Science and Engineering, Jiangsu University of Science and Technology, Zhenjiang 212003, China; 199060030@stu.just.edu.cn (Y.D.); guoyuhang@just.edu.cn (Y.G.); 4School of Electromechanical Engineering, Guangdong University of Technology, Guangzhou 510006, China

**Keywords:** additive manufacturing, Ni based alloy cladding layer, 316L stainless steel, multilayer laser cladding, wear properties

## Abstract

We prepared three kinds of Ni based alloy cladding coatings on 316L stainless steel at different power levels. The microstructure of the cladding layer was observed and analyzed by XRD, metallographic microscope, and SEM. The hardness of the cladding layer was measured, and the wear resistance of it was tested by a friction instrument. The results show that the effect of laser cladding is good, and it has good metallurgical bonding with the substrate. Different microstructures such as dendritic and equiaxed grains can be observed in the cladding layer. With the increase in laser power, more equiaxed and columnar dendrites can be observed. The phase composition of the cladding layer is mainly composed of γ–Ni solid solution and some intermetallic compounds such as Ni_3_B, Cr_5_B_3,_ and Ni_17_Si_3_. The results of EDS show that there are some differences in the distribution of C and Si between dendrites. The hardness of the cladding layer is about 600 HV_0.2_, which is about three times of the substrate (~200 HV_0.2_). Through the analysis of the wear morphology, the substrate wear is serious, there are serious shedding, mainly adhesive wear, and abrasive wear. However, the wear of the cladding layer is slight, which is abrasive wear, and there are some grooves on the surface.

## 1. Introduction

316L stainless steel is a typical low-carbon austenitic stainless steel with good mechanical properties and excellent corrosion resistance, which is widely used in aerospace, chemical, nuclear industry, and marine engineering [1,2,3,4]. However, due to its weak hardness and wear resistance, its application is greatly limited [2]. Therefore, surface strengthening of 316L stainless steel is one of the most important and essential methods to solve this problem, and various technologies for realizing surface modification of materials have entered the field of vision of many researchers.

To date, there have been many methods for surface modification of metal materials, such as surface nanocrystallization [5,6,7], surface carburizing strengthening [8,9], plasma nitriding strengthening [10,11], high current pulsed electron beam (HCPEB) [12,13], spraying [14,15], surface plating [16], and laser additive manufacturing (LAM) [17], etc. However, surface nitriding and carburizing processes are costly and complicated, and may also cause the deposition phase and martensitic transformation to affect the corrosion resistance of stainless steel [18,19]. The spraying process has problems such as poor bonding force between the base material and the coating material, many porosity defects, and a high dilution rate [20]. What’s more, the hard chromium plating (HCP) in common surface plating processes can produce toxic and carcinogenic Cr6 ^+^, which leads to more and more restrictions on its use [21].

As a new manufacturing technology integrating laser, digitization, materials science, and other disciplines, LAM has obtained widespread attention in recent years since it can realize dimension reduction manufacturing, complex forming, and high material utilization [22,23,24]. According to material feed-in methods, LAM can be divided into powder spreading type selective laser melting [25] and powder feeding type laser melting deposition [10,11,26]. Laser melting deposition technology is also called laser cladding (LC), which uses a high-power laser to melt powder and/or wire, and substrate together to obtain a cladding layer with good metallurgical bonding performance [2,27]. Due to the high-quality characteristics of low porosity, low dilution ratio, and crack-free, laser cladding has been widely used to prepare various high-performance composite cladding layers [28]. In addition to improving material performance, laser cladding can also repair material surface defects, which is also the focus of material research in recent years [29,30].

In the selection of cladding layer materials, Ti based, Co based, Fe based, and Ni based alloy powders are widely used. The Ni based alloy can provide higher hardness to improve the wear resistance of the material [31] which is matched with the properties required by 316L stainless steel in different environments. Additionally, the good corrosion resistance provided by the addition of alloying elements (such as Cr) has good compatibility with the application environment of 316L stainless steel [32]. Therefore, the cladding of Ni based alloy on 316L stainless steel has attracted some researchers’ attention. Chang et al. [33] studied the microstructure and mechanical properties of Ni-Cr-Si-B-Fe composite cladding layer prepared by laser cladding. It was found that the cladding layer was mainly composed of amorphous phase and γ-(Fe, Ni) solid solution phase, and the cladding layer showed excellent wear resistance due to the existence of the amorphous phase. Moskal et al. [34] laser cladding the NiCrAlY cladding layer on Inconel625 alloy and 316L stainless steel, and the cladding layer with good metallurgical bonding with the substrate was obtained. Only in the 316L upper cladding layer, strong epitaxial growth was observed. At the same time, the cladding layers of two different substrates had dendrite and interdendritic zone structures rich in different alloy elements. N. Jeyaprakash et al. [4] studied the laser cladding process of cobalt and nickel based hard microlayers on 316L stainless steel substrate, and found that the wear resistance of nickel based cladding was better than that of cobalt cladding and substrate. Wang et al. [1] studied the corrosion behavior of Ni-Cr-Mo laser cladding layer, 316 stainless steel and X70 Steel in a simulated solution of H_2_S and CO_2_. The Ni-Cr-Mo cladding layer has a uniform and dense microstructure, and the morphology does not change after corrosion, showing better corrosion resistance than 316L stainless steel and X70 steel.

Although there are some reports about laser cladding of Ni based alloy on 316L stainless steel, its quantity is relatively small. Therefore, in this study, Ni-Cr-B-Si-C-Fe was selected for multi-layer laser cladding on 316L stainless steel. The phase composition, microstructure evolution, and hardness of the cladding layer were investigated by XRD, SEM, TEM, and microhardness tester. In addition, the wear behavior of laser cladding coating was studied by the ball on the plate tribometer, and the wear mechanism was discussed and analyzed.

## 2. Experimental Procedure

Commercial Ni based alloy (Ni-Cr-B-Si-C-Fe) powder was used as a cladding material. The SEM image of the powder was shown in Figure 1. It could be seen that the shape of the powder was almost equiaxed spherical, and the size was about 50–150 μm, the d_50_ of the powder was 100 μm. The 316L stainless steel, cut into 80 mm × 50 mm × 10 mm (thickness), was applied as the substrate. Before cladding, the substrates were polished with 1000 mesh SiC sandpaper, and sandblasting technique (Al_2_O_3_ ceramic particles) was used over the substrate to have a surface roughness (Ra) of 10 μm. Then, they were soaked in acetone to remove oil, washed with alcohol, and dried to keep the surface clean. The chemical composition of 316L stainless steel and Ni based alloy powder were shown in Table 1.

A RFL-C3300 (Raycus, Wuhan) high power continuous fiber laser device was used in the laser cladding test. DPSF-2 powder feeder was used to feed powder synchronously in an oblique direction, and argon was used as a power source to accurately send powder to the laser spot. During the cladding process, argon was used to protect the molten pool to avoid oxidation. The three different configurations, named LC1, LC2, and LC3, were used for the cladding experiment. The detailed parameters were listed in Table 2. The cladding layer was divided into three layers. The first layer and the third layer were along the length direction of the plate, and the second layer was along the width direction of the plate. The samples produced by cladding and their corresponding macro photographs were shown in Figure 2. It can be seen from the figure that there are some fluctuations in the thickness of each layer due to the influence of different tracks. The first layer of sample LC1 was about 800–1400 μm, the second layer was about 300–1000 μm, and the third layer was relatively thick, about 1600–1800 μm; the thickness of each layer of the other two samples was similar to that of sample LC1.

After cladding, the samples were cut, cleaned, and degreased with acetone alcohol, and then a standard metallographic sample preparation process (grinding, polishing, washing, and drying) was carried out to obtain mirror-surface metallographic samples. The phase identification of the cladding layer was performed by X-ray diffractometer (XRD) with a Cu-k α radiation at a voltage of 40 kV. The micromorphology of the cladding layer was observed by JSM-6480 tungsten filament scanning electron microscope (SEM) and ZEISS Merlin compact field-emission scanning electron microscope (FE-SEM), and the element content in the micro area was measured by their equipped energy dispersive spectrometer (EDS), meanwhile, transmission electron microscope (TEM) was also used to observe the microstructure. A KB-30S-FA Automatic Vickers Hardness Tester was used to test the hardness of the cladding layer and the substrate with the applied pressure of 0.2 kg and the holding time of 15 s. The dry sliding ball-on-plate wear behavior of the cladding layer and the substrate were studied by UMT-2 high-temperature friction and wear tester. For the dry sliding friction test, the sliding time was 1800 s with a sliding friction load of 30 N, the relative speed between coating and counter-body was 10 mm/s, and the counter-body ball was an Al_2_O_3_ ceramic ball with a diameter of 10 mm. Before the wear test, the surface roughness (Sq) of the substrate, LC1, LC2, and LC3 were 0.012, 0.022, 0.022, and 0.027 μm, respectively. Finally, the three-dimensional morphology of wear marks was observed by LEXTOLS4000 confocal laser scanning microscope, and the microwear morphology was observed through SEM.

## 3. Results and Discussion

### 3.1. Microstructure Observation and Analysis

The process of the first cladding layer in the three samples is the same. The microstructure of the interface between the substrate and the cladding layer in LC1 is observed, and the SEM image is shown in Figure 3. It can be seen that the cladding layer has good metallurgical bonding with the matrix material, and some small pores are randomly distributed in the cladding layer. Planar and cellular crystals can be observed at the substrate cladding interface, and similar structures have been observed in Li et al. [35]. On multilayer laser cladding of 308L stainless steel, which is attributed to the “flat interface growth mode.” The two main parameters can be used to explain the microstructure evolution, i.e., temperature gradient G and growth rate R. The change of G–R ratio will change the solidification mode, resulting in the change of microstructure morphology and size [36]. At a large G–R ratio, planar crystals are formed, and with the decrease in the G–R ratio, cellular crystals are formed. There is no heat accumulation during the first layer cladding. At this time, the maximum value of G and the minimum value of R appears at the bottom of the molten pool, and the planar crystals are formed. With the decrease in the G–R ratio, cellular crystals appear between the planar crystals and the liquid phase. However, it can be seen that such morphology is small-scale, only within 10–20 μm above the interface.

Figure 4 shows the typical OM diagram of the second cladding layer of LC1. It can be seen from the figure that there are two equiaxed crystal bands at the top and bottom, which are called bonding zones. As shown in Figure 4b–e, the enlarged structure of the lower bonding zone and the upper bonding zone, respectively, show that the grains in the bonding zone are equiaxed (Figure 4b,e), and the banding zone between the second and third layer has a wider scale. In the middle of the second cladding layer, columnar dendrites and some equiaxed dendrites are obvious (Figure 4c,d).

The formation of the bonding zone is the result of reheating at the top of the early and latter layers. During multiple cladding, the top of the early layer is melted, and the area below the top is reheated, resulting in grains growth in this area [37]. When it comes to columnar dendrites and equiaxed dendrites, they can still be described by the change of the G–R ratio. After cellular crystals appear, columnar dendrites are formed with a further decrease in the G–R ratio. At a low G–R ratio, columnar dendrites transform into equiaxed dendrites (CET) and equiaxed dendrites are formed [36,37]. There is no planar crystal and cellular crystal at the bottom of the second cladding layer, which is due to the rapid solidification rate, which does not have enough time to stably grow into planar crystals and cellular crystals [38]. Furthermore, the heat accumulation during the second layer cladding is also a factor that has to be considered. Due to the heat accumulation, the temperature gradient at the bottom of the molten pool decreases, resulting in the decrease in the G–R ratio, which is easier to form columnar dendrites. The closer to the middle of the molten pool, the smaller the G–R ratio is, and equiaxed dendrites are formed due to CET finally. In addition, some oblique growing dendrites (steering dendrites) were observed below the bonding zone, which generally appeared near the top of the molten pool, which was caused by the movement of the laser beam and the contact with air, resulting in the change of the direction of temperature gradient; at the same time, the flow in the molten pool also caused the dendrite to deflect during the growth process.

The microstructure and morphology of the interface of the second and third layers of the three samples are similar to the above analysis, but there are still some differences (Figure 5). In LC2 and LC3, there is a large area of equiaxed dendrites under the bonding zone. With the increase in laser power and a cladding layer, the heat accumulation increases, the G–R ratio decreases, and more equiaxed dendrites are formed. On the other hand, the smaller G–R ratio caused by more heat accumulation will also enable more columnar dendrites to be observed. Meanwhile, the more complex temperature gradient will make the directional distribution of columnar crystals more complex.

### 3.2. Phase Composition and Analysis

XRD diffraction was used to analyze the phase composition of Ni based alloy powder and a cladding layer, and the results are shown in Figure 6. It can be seen from the figure that the main phases in the alloy powder are γ–Ni, Ni_3_B, Cr_5_B_3,_ and Ni_17_Si_3_. After laser cladding, it is obvious that many small peaks disappear, but there is no significant difference in the phase of the three cladding layers under different power. However, there is the main peak at 51–52° in sample LC1, and the peak at the corresponding position is a secondary strong peak in both powder and LC2 and LC3, which may be due to the preferred orientation in the sample structure under the cladding condition. For face-centered cubic structure, the preferred growth direction is <100> orientation group. However, in a different laser cladding processes, the complex temperature gradient, heat dissipation direction, and solidification speed will affect the crystal growth. Finally, the crystal will choose the direction with the smallest angle between the solid-liquid interface front velocity and the front velocity. Additionally, the γ–Ni peak shifts to the left in the cladding layer, which is attributed to the lattice distortion caused by the solid solution of more alloy elements into the γ–Ni solid solution during rapid melting and cooling [39]. It can be seen that the main phases in the cladding layer are γ–Ni solid solution, Ni_3_B, Cr_5_B_3,_ and Ni_17_Si_3_. Among them, borides such as Ni_3_B and Cr_5_B_3_ can provide good hardness and wear resistance for the cladding layer. The element C has not been determined due to its low content.

The microstructure of the cladding layers was observed and analyzed by SEM, as shown in Figure 7, the SEM pictures of the middle regions of the second layer of three kinds of samples were shown. It can be seen that sample LC2 has coarser dendrites than sample LC1 and sample LC3, this is because when the input power increases, the corresponding heat input will also increase, and the solidification rate will decrease, which is conducive to the growth of dendrite. However, with the further increase in the input power, the heat input continues to increase, and the heat accumulation increases. When the G–R ratio is low, it is easier to form equiaxed dendrites, and the dense equiaxed dendrites restrict each other during the growth process, which makes the grains smaller.

Also, the microstructure near the surface of the third layer was observed, as shown in Figure 8. In the structure near the outer surface of LC1, no obvious dendrites morphology was observed, but a strip-shaped phase and honeycomb structure could be seen. The elements in the strip phase were analyzed by EDS in Figure 9. the results show that the strip phase is rich in Ni, Si, and Fe elements. However, the dendrite morphology can still be observed in LC2 and LC3, and the network structure can also be observed in the interdendritic zone. Such a network structure can also be observed in Figure 7a,b. EDS is used to analyze the elements of the network structure, and the results are shown in Figure 9. FE-SEM was used to observe the microstructure of Figure 8a–c in more detail. It can be seen from Figure 8d–f that the honeycomb structure in sample LC1 is composed of fine grains. In the study of [40], it is believed that such fine grains can provide higher hardness. The most obvious dendrite structure can be observed in LC3, and the network structure is distributed among the dendrites; compared with the finer microstructure, the coarse dendrite structure provides lower hardness.

According to the EDS results (Figure 9), the element C is mainly measured in the dendrites and also distributed in the interdendritic region, but it is difficult to detect in the interdendritic network structure. Si element is detected in a higher content in the network structure, combined with XRD results, it can be inferred that the Ni_17_Si_3_ phase mainly exists in the network structure. However, a higher content of Si can also be detected in the third layer of sample LC1, which forms a long strip structure with Ni and Fe elements. Additionally, the C element can be detected in a higher content locally, or it is due to the formation of carbides during the cladding process. Figure 10 shows the TEM morphology and the selected area electron diffraction pattern (SAED) of the two regions. From Figure 10a, the black phases can be observed, and the SAED in region A indicates the γ–Ni phase. In region B, the SAED of the black phases is identified as (Ni, Cr)_3_C_2_ carbide whose size is about 0.5–2 μm. This indicates that the locally high concentration of C element detected in the EDS results is indeed carbide. As for the light element B, EDS has not been detected, but in the study of laser cladding Ni-Cr-B-Si-C on 316L SS by Jeyaprakash [4], it is reported that Ni_3_B and γ–Ni (Fe, Si, Cr) form interdendritic network eutectic structure.

### 3.3. Microhardness and Wear Behavior

The microhardness of the cross section of the substrate and cladding layer was tested, as shown in Figure 11. As can be seen from it, the microhardness of the 316L substrate is about 200 HV_0.2_, and the hardness of the cladding layer is significantly higher than that of the substrate. At the same time, it can be seen from the figure that the microhardness of the substrate decreases slightly near the junction of the cladding layer and the substrate, which is due to the influence of temperature on the substrate during the cladding process. The microhardness of the cladding layer rises abruptly to about 600 HV_0.2_ after crossing the interface. LC1 has the finest grains near the outer surface, which contributes to the least fluctuation of hardness, followed by LC2 and LC3. According to the microhardness distribution curve of the cladding layer section, the microhardness of the inner region is slightly lower than that of the outer region, which is due to the coarser microstructure of the inner region. Figure 11b shows the average hardness of the three cladding layers. It can be seen that the average hardness of the cladding layer can reach about 2.8–3.0 times of the substrate. In addition to the formation of residual stress during rapid cooling, the high hardness of the cladding layer is attributed to the formation of a γ–Ni solid solution formed by solid solution of alloy elements and the formation of intermetallic compounds Ni_17_Si_3_, Ni_3_B, and Cr_5_B_3_. With the increase in laser power, the average microhardness of the cladding layer decreases, but the decreasing range gradually decreases. The maximum average hardness is LC1 (~605 HV_0.2_), which is due to the higher hardness contributed by finer microstructure. Then LC2 (~568 HV_0.2_) and minimum LC3 (~563 HV_0.2_).

Since the tribological properties are not the inherent properties of materials but depend on the mechanical properties, roughness, friction, and other factors [41,42], the substrate and three cladding layers are polished under the same conditions, and the same experimental conditions are used to reduce the error of four samples in the friction test. Friction and wear experiments were carried out on the surface of the substrate and multi-layer cladding coating. Figure 12a,b shows the friction coefficient curve and average friction coefficient of the substrate and three kinds of samples, respectively. It can be seen from the curve that the friction coefficient of the substrate keeps a small fluctuation from the beginning to the end. This is due to the low C content in the microstructure of 316L stainless steel, which is mainly composed of austenite and a certain amount of ferrite, and there is no other hard and brittle phase in the soft structure, so the friction coefficient changes little during the friction process. Additionally, the friction coefficient of the matrix has a tendency to decrease gradually (analyzed in the subsequent wear track morphology). However, the friction coefficients of the three cladding layers fluctuate greatly, which is due to the effect of debris falling off during the friction process; the fluctuation of LC1 is the most obvious (between 0.31 and 0.48), especially after the peak value, it drops sharply. The friction coefficient of LC2 and LC3 is more gentle than that of LC1, and there is no abrupt drop after the peak value. The fluctuation range of LC2 wear coefficient is 0.28 to 0.42, and LC3 is 0.33 to 0.42. It can be seen from Figure 11b that the average friction coefficients of the matrix, LC1, LC2, and LC3 are 0.53 ± 0.02, 0.37 ± 0.03, 0.35 ± 0.02, and 0.36 ± 0.02, respectively, and the friction coefficient of the matrix is about 1.5 times of that of three cladding layers.

The three-dimensional morphology and the wear profile of the substrate and cladding layer are shown in Figure 13. It can be seen from the three-dimensional wear map of the substrate that there are a large number of protrusions and pits. The surface of the wear tracks is uneven, and the material is extruded and piled up on both sides, showing a ridge shape (Figure 13a). In contrast, the scratch of cladding layers is smoother, and it is difficult to see uneven protrusion on the surface of wear tracks of cladding layers, but there are some gullies and ridges. It can be seen from the contour curves of wear tracks (Figure 13b), that the wear track width of the substrate is ~2000 μm, which is much larger than ~450 μm of LC1, ~460 μm of LC2, and ~440 μm of LC3. The maximum wear track depth of the substrate is ~38 μm, which is still far greater than ~7 μm of LC1, ~7 μm of LC2 and ~9 μm of LC3 (inset in Figure 13b). After laser cladding, the surface of 316L stainless steel can be effectively protected.

Figure 14 shows the SEM images of wear tracks of substrate and cladding layers. As can be seen from Figure 14a, large grooves can be observed on the worn surface of the substrate. The appearance of such plow marks represents high plastic deformation and low wear resistance [43]. Due to the forced transfer of materials by plowing, stacking and delamination of materials, as well as corresponding drop pits and spalling, can be observed. Small steps of transverse deformation layer formed by material transfer and flaking soft material influence friction process continuously. The relatively soft substrate, in the process of friction, can produce new shedding materials, in the repeated wear of the friction ball back and forth, in addition to adhering to the wear surface, it can be used as an abrasive to intensify the wear of substrate. With the increase in the number of cycles, the deformation layer can be transformed into a protective film to reduce the friction [44], which leads to the gradual decrease in the friction coefficient. The corresponding phenomenon can be seen in Figure 11. However, this protection is negligible relative to wear, and the base material is still being worn. Generally speaking, the wear mechanism of the substrate is abrasive wear and adhesive wear.

Figure 14b shows the wear morphology of the LC1 cladding layer. It can be seen that there is also a material stacking phenomenon in the wear morphology of the cladding layer, which is only longitudinal distribution, consistent with the wear direction, and there is no mass transfer of materials like the matrix. The appearance of the ridge is caused by a small amount of material having to be extruded to both sides under the action of vertical stress and shear stress in the wear process, corresponding to the ridge dent morphology of the worn surface in the three-dimensional diagram (Figure 12). In addition, pits of different sizes due to peeling off, as well as small wear debris and mild groove, can be seen on the worn surface of LC1. This kind of not deep groove is mainly caused by the abrasive wear caused by small wear debris, which also represents that the cladding layer has good wear resistance. The same ridge morphology, fine wear debris, and slight grooves can also be seen in the wear morphology of LC2 and LC3 cladding layers (Figure 14c,d), indicating that the three coatings have similar good wear resistance. However, there are no large drop pits in LC2 and LC3 as shown in Figure 14b. Combined with the microstructure of the LC1 cladding layer, it can be referred to that there is a long strip phase in the outer surface structure of LC1. Due to the large aspect ratio, this long strip phase is easy to break and peel off from the cladding layer under the action of external stress. The material is dug out from the cladding layer by the falling debris in the process of continuous cyclic wear, resulting in large spalling pits. At the same time, this behavior will lead to an increase in friction, which will lead to an increase in friction coefficient. therefore, the average friction coefficient of LC1 is slightly higher than that of LC2 and LC3, although the friction coefficient generally decreases with the increase in hardness [45]. In sample LC1, fine dendrites can provide higher hardness and good wear resistance, but the appearance of long strip phases will lead to reduced wear performance. In the sample LC2, the dendrites on the surface are slightly coarse, resulting in a slight decrease in hardness and wear resistance. Coarse dendrites can be observed in LC3, and the corresponding hardness and wear resistance also decreases. However, no long strip phases appeared in the two samples, which ensures that their wear resistance is not affected. In a word, the main wear mechanism of the cladding layer is abrasive wear. From the point of view of wear, the Ni based alloy coating can provide good protection for 316L stainless steel substrate, to avoid failure or damage due to a large number of wears in the corresponding environment, and effectively improve the wear life of the material. Moreover, the sample LC2 has the lowest wear coefficient, combined with the SEM images of the wear track, it can also be seen that its wear is good, so its wear resistance is better than the other two samples.

## 4. Conclusions

The bonding zone between cladding layers is observed, and the microstructure is coarse dendrite formed by reheating. The planar and cellular crystals observed at the substrate cladding interface were not observed in the subsequent cladding layer due to heat accumulation. More equiaxed dendrites can be observed in the cladding layer, and more coarse columnar dendrites can be observed in the bonding zone.There is no significant difference in the composition of the three kinds of cladding layers, and the main phases are γ–Ni, Ni_3_B, Cr_5_B_3,_ and Ni_17_Si_3_. EDS results show that C element mainly exists in the dendrite, while Si element mainly distributes in the network structure and dendrite between the dendrites, and there is no obvious difference in the distribution of other elements. Besides, only Ni, Si, and Fe were detected in the long strip phase near the outer surface of LC1.The average hardness of the cladding layer is about 2.8–3.0 times that of the substrate, which can be attributed to the formation of γ–Ni solid solution and intermetallic compounds Ni_17_Si_3_, Ni_3_B, and Cr_5_B_3_. The microhardness of the inner region is slightly lower than that of the outer region, and the hardness of LC1 is higher than that of LC2 and LC3, which can be explained according to the microstructure.The friction coefficient of the substrate is about 1.5 times that of the cladding layer. The wear mechanism of the substrate is mainly abrasive wear and adhesive wear, while the main wear mechanism of the cladding layer is abrasive wear. The cladding layer can provide good protection, and the sample LC2 performs best.

## Figures and Tables

**Figure 1 materials-14-00781-f001:**
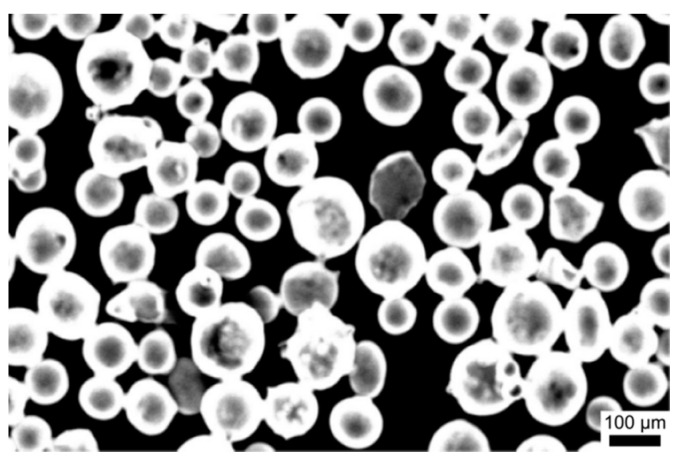
The morphology of Ni-base alloy powders.

**Figure 2 materials-14-00781-f002:**
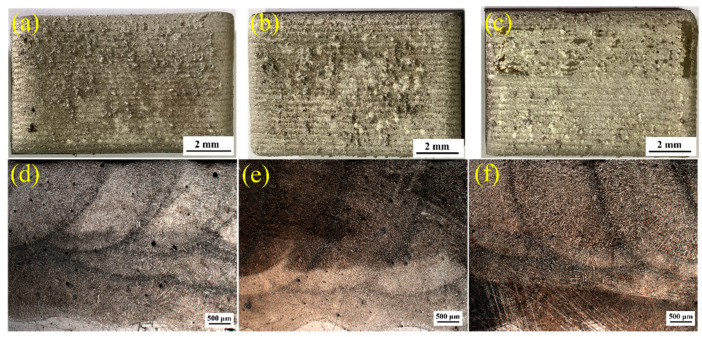
Macroscopic morphologies of three cladding coatings (**a**,**d**) sample LC1; (**b**,**e**) sample LC2; (**c**,**f**) sample LC3.

**Figure 3 materials-14-00781-f003:**
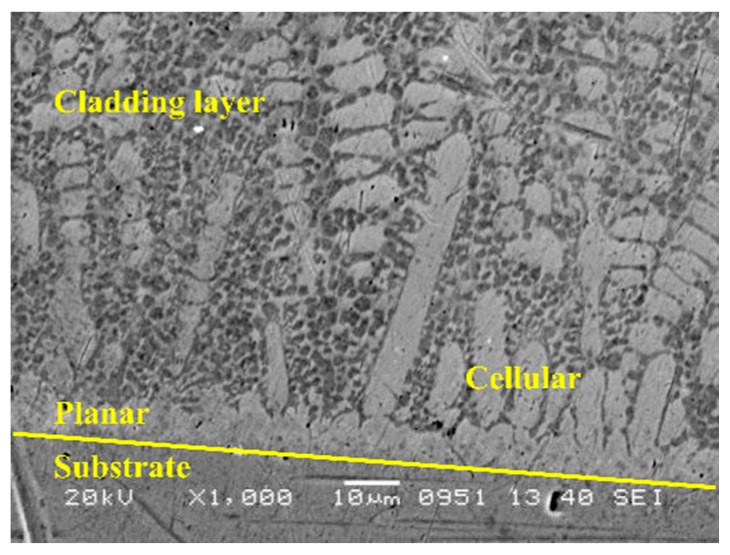
Typical microstructure at the junction between 316L SS substrate and coating in sample LC1.

**Figure 4 materials-14-00781-f004:**
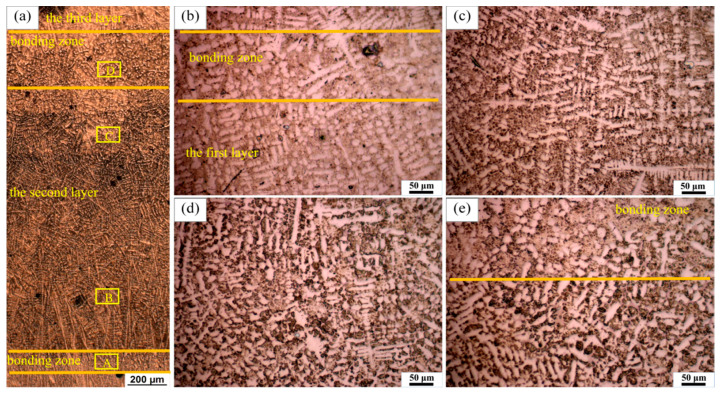
A typical microstructure of the second cladding in sample LC1: (**b****–e**) correspond to A–D in (**a**), respectively.

**Figure 5 materials-14-00781-f005:**
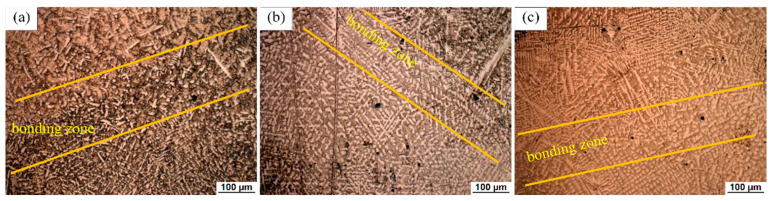
The microstructure of the bonding zone between the second and third layers in (**a**) sample LC1; (**b**) LC2; (**c**) LC3.

**Figure 6 materials-14-00781-f006:**
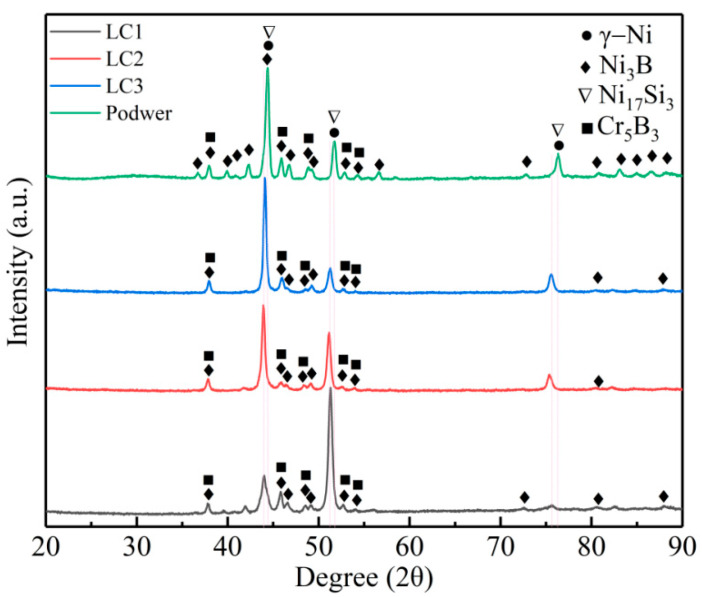
XRD patterns of powders and three cladding coatings.

**Figure 7 materials-14-00781-f007:**
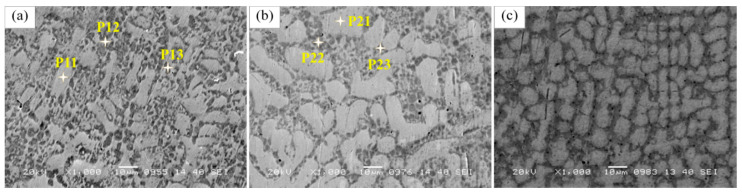
SEM images of the middle region in the second cladding coating: (**a**) sample LC1; (**b**) LC2; (**c**) LC3.

**Figure 8 materials-14-00781-f008:**
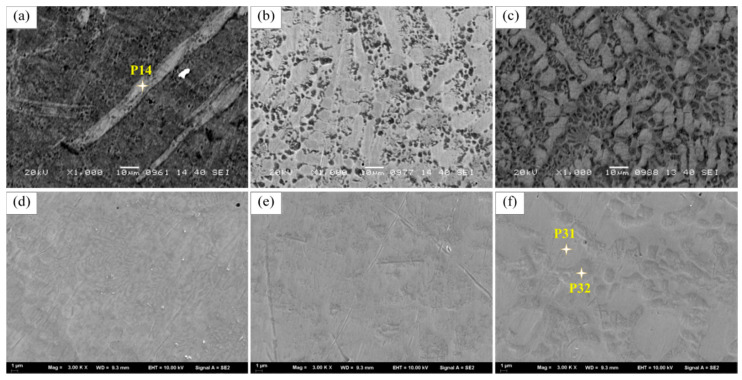
SEM and FE-SEM images of the surface region in the third cladding coatings: (**a**,**d**) sample LC1; (**b**,**e**) LC2; (**c**,**f**) LC3.

**Figure 9 materials-14-00781-f009:**
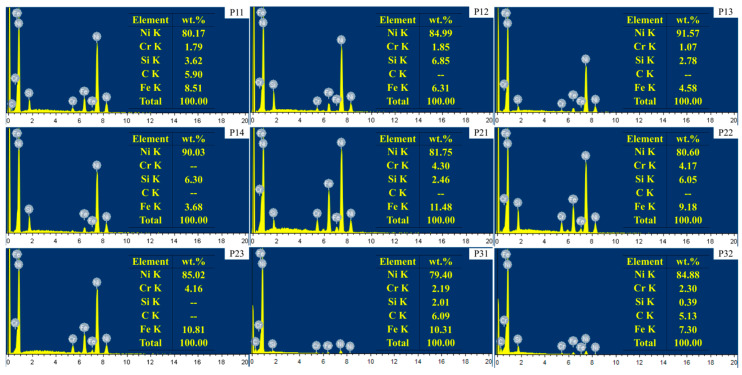
The Corresponding EDS results in Figure 7 and Figure 8.

**Figure 10 materials-14-00781-f010:**
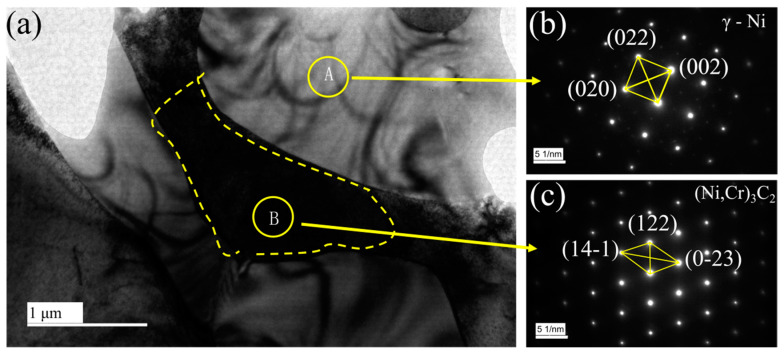
TEM image and SAED patterns of the coating. (**a**) TEM image, (**b**) SAED patterns of region A, (**c**) SAED patterns of region B.

**Figure 11 materials-14-00781-f011:**
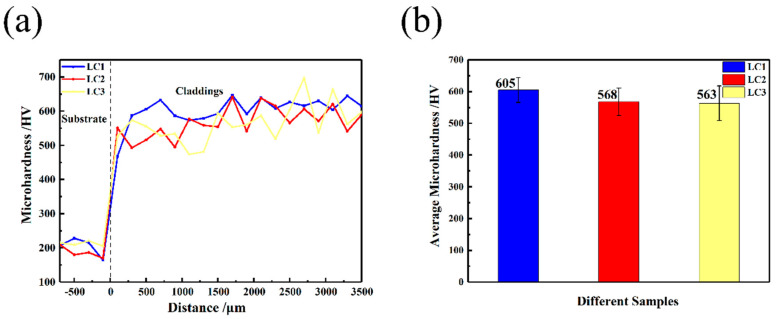
Vickers microhardness of the substrate and cladding coatings: (**a**) Microhardness distribution of cross section; (**b**) average microhardness.

**Figure 12 materials-14-00781-f012:**
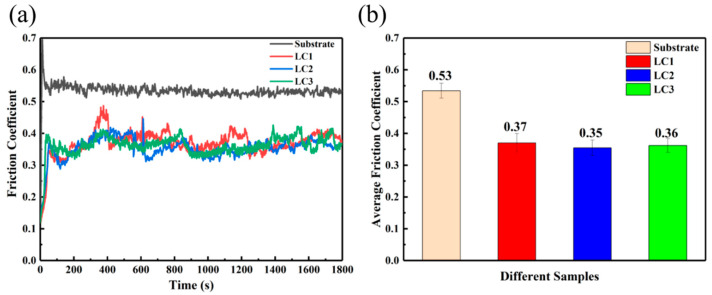
The friction coefficient of the substrate and cladding coatings: (**a**) friction coefficient curves; (**b**) average friction coefficient.

**Figure 13 materials-14-00781-f013:**
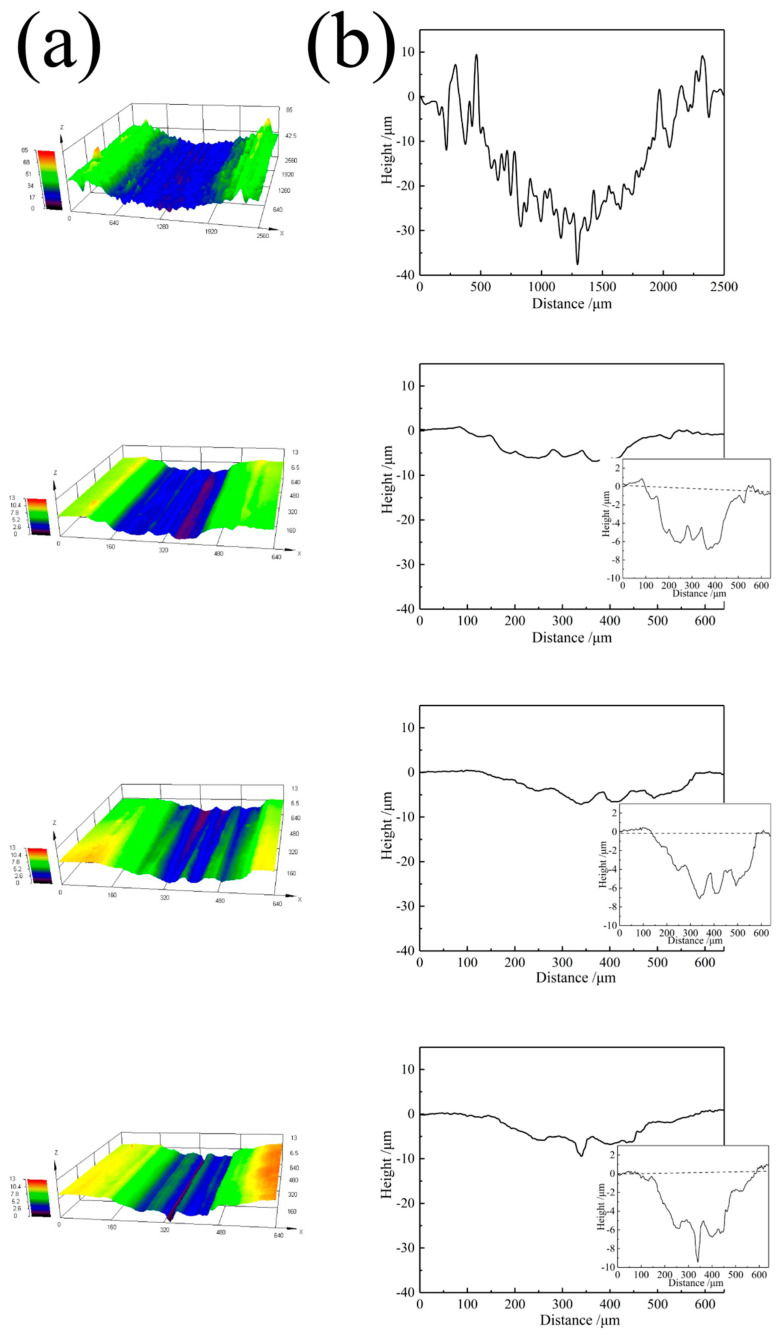
3D morphology (**a**) and wear profiles (**b**) of the substrate and cladding coatings.

**Figure 14 materials-14-00781-f014:**
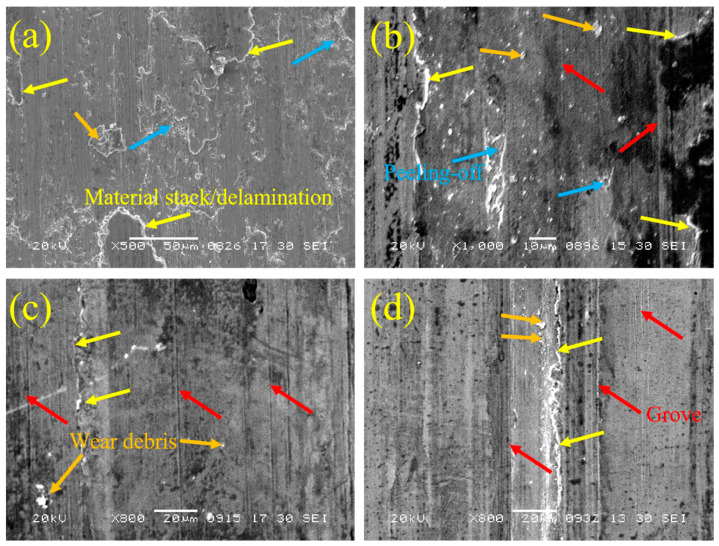
SEM images of worn surface morphology: (**a**) substrate; (**b**) sample LC1; (**c**) sample LC2; (**d**) sample LC3.

**Table 1 materials-14-00781-t001:** Chemical composition of 316L stainless steel and powder (wt%).

Materials	C	P	Cr	S	Mn	Mo	B	Si	Ni	Fe
316L SS	0.023	0.034	16.4	0.57	1.37	2.16	--	0.69	10.03	Bal.
Nickel alloy	0.03	--	6.00	--	--	--	3.00	1.50	Bal.	0.38

**Table 2 materials-14-00781-t002:** Different parameters of laser cladding.

Samples	Laser Power (kW)		Powder Feed Rate (g/min)	Scanning Velocity (mm/s)	Spot Diameter (mm)	Gas Flow Rate (L/min)	Overlap Rate
	1st Layer	2nd and 3rd					
LC1	1.8	2.0	30	5	4	15	50%
LC2	1.8	2.2	30	5	4	15	50%
LC3	1.8	2.4	30	5	4	15	50%

## Data Availability

The data that support the findings of this study are available on request from the corresponding author.

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
