# Peer review of "Microstructure and Wear Resistance of Multi-Layer Ni-Based Alloy Cladding Coating on 316L SS under Different Laser Power"

_materials, 2021, doi:10.3390/ma14040781_

Round 1
Reviewer 1 Report
1. Please add the particle size distribution or at least d50 value.
2. Surface roughness after sand-blasting was measured by which roughness parameter?
3. Please add macrophotography of hardfacing layers.
4. Please add information about the material of counter-body for sliding test.
5. Please add quantitative analysis for XRD investigations.
6. How many indentations have been carried out for microhardness determination?
7. Please change "wear scar" into "wear track".
8. At last, but not least - Authors did not determinate wear resistance of the cladded layers.
Author Response
Dear Reviewer,
We greatly appreciate the comments and suggestions made by the reviewers. Those are quite reasonable and constructive. Accordingly, our paper has been thoroughly revised.
Sincerely,
Qian Shaoxiang

Reviewer 2 Report
The work concerns the interesting topic of increasing the abrasion resistance of 316L steel. While reading the work, many questions are asked, which are formulated below.
What was the premise for the use of a three-layer, non-two-layer coating?
It seems useful to mention the total thickness of each layer produced. This thickness can only be guessed from Figs 4 and 10. But did all cladding coatings actually have the same thickness? Cladding coatings were assessed for abrasion resistance, and the depths of abrasions did not exceed the thickness of the third, outer layer. For this reason, the most important is the characteristics of the third layer, along with the visualization of the surface condition and roughness after cllading. Roughness affects the coefficient of friction and affects the depth of rubs. As shown in Figure 8, each of the cladding coatings LC1, LC2 and LC3 has a different microstructure despite similar diffraction patterns. One gets the impression that the description of these microstructures is not sufficient and is based mainly on the cited works, and not on own research. Firstly, the presence of the Ni17Si3 phase on the diffractograms, as well as the Cr5B3 phases, was declared, but not identified because the reflections overlap. Secondly, the EDS analysis of the third layer is sketchy - you can have doubts because the Authors write "Si is mainly distributed in the network structure between dendrites, less in dendrites, and a little or no Si in interdendritic." while in point P14 the Si concentration is one of the highest. Additionally, no reference was made to the local concentration of C - could it be carbides? In conclusion, the relationship between the microstructure of the third layer and the abrasion resistance should be demonstrated. It would also be good to suggest some conclusion about the abrasion resistance for a longer abrasion time that would include the second layer.
It is not known what benefits result from the presentation of FEM-SEM, because the „It can be seen from Figure8 (d) ~ (f) that the honeycomb structure in sample LC1 is actually very fine" what the authors write about is hardly visible.
The authors tried to describe the micrographs in detail, so it could be suggested to supplement micrograph No. 3 with explanations - indicate planar and cellular crystals.
Author Response

(The authors gave the same response as above.)

Reviewer 3 Report
The author presents a manuscript base on the relevance of the protective coating of 316L SS applied by laser cladding. The objective of this research work is to evaluate the influence of laser power over the wear properties.
I would recommend this manuscript for publication, after a deep revision of the comments and query raised.
Introduction.
Line 62: The yield strength of bulk material is hardly improved by adding a cladding layer. Please consider removing this phrase.
Line 68: Replace si by Si
Experimental
Line 102: The author should say "configuration" instead of parameters since only one parameter was altered.
Line 108: Figure 2. Please incorporate a micro mark as a guideline of the sample dimension.
Results and discussion
Line 268: Figure 10. Could the author provide the interlayer position within this graph?. Or only the microhardness was evaluated within the first layer of cladding?
Line 287: “friction coefficients of matrix, LC1, LC2 and LC3 are 0.6316, 0.3702, 0.3544 and 0.3619, respectively”, Author must change this number. If a Friction coefficient (FC) of 0.6316 is given, this means you can determine an FC of 0.6315
Figure 11 mentions 0.5342 to the FC of the matrix. Please check your results.
A valid FC could be 0.37 for LC1 sample, and please give the maximum and lower values registered.
Line 303: Figure 12. Could the author provide the overall thickness of the cladding? It seems you are only testing the 3er layer, if so. What is the reason for the 3 consecutive layers?
Author Response

(The authors gave the same response as above.)

Reviewer 4 Report
This paper is mainly investigated that the analysis of phase composition and wear resistance of Ni-based cladding coating with 316L SS steel after different laser power processing. The authors pointed out the phase composition concluded γ-Ni solid solution and some intermetallic compounds. They also confirmed that the wear resistance of the cladding layer, especially the abrasive wear, got improved. It indeed indicated that there are lots of works about this research. Although, there are some unperfects occurring in this paper. Worthy to notice, the expression of abstract is partial to Chinese and there is no light spot to interest reviewers. Please think twice about abstract writing.
(1) In the first part, why are you stating two times surface strengthening methods with a different conclusion? If you just want to express the laser cladding and the other methods, please state it just once and give more details about the reasons for the wiled application of laser cladding.
(2) The authors should improve the Introduction, since there are no other methods of changing the surface properties for different methods of modification. Authors can use the following articles for analysis: https://doi.org/10.3390/nano10122398, https://doi.org/10.1016/j.jmrt.2020.06.016, https://doi.org/10.1016/j.vacuum.2019.06.022
(3) Why is multilayer not the single layer processed on the 316 stainless steel? If the multilayer with more metal material and thick modification layer has a high cost and time-consuming?
(4) it is better to mark those two parts in fig 3, which part is the cladding layer and which part is base metal.
(5) If the results summarised in the other’s reports are consistent with it in this research?
(6) the word “ of course” should not be used in research.
(7) Please revise all specialized words in the third part!
(8) In fig 5(a), there is a visible boundary, if it is the bonding zone in the slashing area?
(9) there some logical mistakes in some sentences.
(10) In Fig 8. there is no annotation for (d), (e) and (f)
(11) the word “small” is used to describe the grain size, not the microstructure.
(12) the measuring place of all samples is the top layer of modification layers. Where is the inner region and outer region?
Author Response

(The authors gave the same response as above.)

Round 2
Reviewer 1 Report
All my remarks have been included in the revised version.
Author Response
Dear Reviewer
Thanks again for the comments and suggestions of the reviewer.
Sincerely,
Qian Shaoxiang
Ph. D.
Jiangsu University
NO.301 Xuefu Road, Zhenjiang, JiangSu Province, CHINA
Tel.: + 86 511-88962013
E-mail: qiansx@zjc.edu.cn
Reviewer 3 Report
Congratulation on your revision, I look forward to accepting your paper for publication.
The authors have provided an improved version of their manuscript.
The questions and points raised during reviewing were resolved completely. The new revised figure 2 gives important information.
Please give a deep revision of the English language, including figure captions.
Author Response

(The authors gave the same response as above.)

Reviewer 4 Report
Authors made corrections according my comments. No new comments.
Author Response

(The authors gave the same response as above.)
